# Advances in Psoriasis Research: Decoding Immune Circuits and Developing Novel Therapies

**DOI:** 10.3390/ijms26189233

**Published:** 2025-09-21

**Authors:** Lanying Wang, Ruiling Liu, Yulu Tang, Yuanfang Ma, Guimei Wang, Qingguo Ruan, Shijun J. Zheng

**Affiliations:** 1Joint National Laboratory for Antibody Drug Engineering, Henan University School of Medicine, Kaifeng 475004, China; lan131600@163.com (L.W.); ruilingliu@henu.edu.cn (R.L.); 18238290585@163.com (Y.T.); mayf@henu.edu.cn (Y.M.); 17726067138@163.com (G.W.); 2School of Basic Medical Sciences, Henan University School of Medicine, Kaifeng 475004, China

**Keywords:** psoriasis, IL-23/IL-17 axis, biologicals, targeted therapy, CAR-T cell therapy

## Abstract

Psoriasis is a chronic inflammatory autoimmune skin disease characterized by erythematous plaques covered with silvery-white scales, often accompanied by systemic complications such as psoriatic arthritis and cardiovascular diseases. The disease and its systemic complications substantially impair quality of life, compromise socioeconomic status, and threaten patient safety. The occurrence and progression of this disease are related to the IL-23/IL-17 axis and involve the aberrant activation and interactions of multiple immune cells, along with genetic predispositions and environmental triggers. Although current therapeutic approaches, including topical agents, systemic medications, biologic agents targeting key cytokines, and Janus Kinase inhibitors, can control symptoms and delay disease progression, a complete cure has not been achieved. Furthermore, these strategies face challenges relating to the cost, safety, efficacy and precision of targeting. This review summarizes recent advances in mechanistic research, highlighting the interplay among microorganisms, innate and adaptive immunity in psoriasis. We also evaluate a range of emerging therapies, including biologics, small-molecule inhibitors, Chimeric antigen receptor T-cell cell therapy, RNA interference-based strategies, and alternative medicine. Specifically, we focus on their novel mechanisms, efficacy challenges, safety profiles, and targeting accuracy. Finally, we assess their potential in personalized treatment, aiming to achieve long-term remission, and propose the future prospects of precision medicine in psoriasis management.

## 1. Introduction

The erroneous self-attack and destruction of tissue or organ by the immune system can result in autoimmune diseases. To date, over 150 distinct autoimmune disorders have been documented, including psoriasis, type 1 diabetes mellitus, multiple sclerosis, systemic lupus erythematosus, rheumatoid arthritis (RA), and so on [1]. Psoriasis is a persistent, recurrent, immune-mediated inflammatory dermatosis that can affect individuals across diverse geographic regions, ethnicities, and age groups globally [2]. Psoriasis typically presents clinically as well-demarcated erythematous plaques or papules covered with characteristic silvery-white scales, frequently accompanied by marked pruritus, pain, and other distressing symptoms. As a systemic inflammatory disorder, psoriasis manifests not only as cutaneous lesions but is also associated with multiple comorbidities, including psoriatic arthritis [3], metabolic syndrome [4,5] and depression [6].

The pathogenesis of psoriasis involves multiple cell types and cytokines as well as both genetic predisposition and environmental triggers. Current evidence indicates that the interleukin (IL)-23/IL-17 axis is the central driver of psoriatic inflammation [7,8,9]. In addition, genetic factors are also important in the development of psoriasis [10]. Environmental factors such as infections, stress, smoking, and climate changes are also considered important triggers of psoriasis [11,12]. Furthermore, lifestyle factors such as psychological stress and sleep deprivation can exacerbate psoriasis symptoms by affecting the immune system and skin barrier function.

Currently, the treatment strategies for psoriasis primarily include topical therapy, systemic therapy, and biologic therapy. Conventional topical treatments commonly involve corticosteroids and vitamin D3 analogs. Conventional systemic therapies encompass phototherapy, methotrexate, and cyclosporine. Recent advances in biologic agents, particularly monoclonal antibodies against IL-17 and IL-23, have demonstrated significant clinical efficacy and a favorable safety profile in psoriasis treatment. However, their long-term use may be associated with increased risks of drug resistance and infections [13,14]. Moreover, the advent of innovative cellular therapies, such as Chimeric antigen receptor T-cell therapy, hold much promise for the treatment and control of psoriasis. This review updates recent advances in psoriasis pathogenesis, highlights the most promising therapeutic strategies, and offers new insights for future individualized treatment.

## 2. Epidemiology and Disease Burden

Psoriasis is a globally distributed disease [15]. The worldwide prevalence of psoriasis exhibits significant geographical variation, ranging from 11.4% in Norway [16] to 0.09% in the United Republic of Tanzania [17]. The varied prevalence of psoriasis might be due to heterogeneous diagnostic criteria and the absence of mandatory registration systems across countries. It was reported that the global prevalence of psoriasis is increasing [18,19], which might be associated with the misuse of antimicrobial drugs and environmental pollution [20,21]. Furthermore, an UK-based study revealed that individuals with elevated genetic susceptibility who were exposed to nitrogen dioxide (NO_2_), nitrogen oxides (NO_x_), and particular materials under 10 µm (PM_10_) showed significantly increased risk of psoriasis development [22]. However, the biological mechanisms by which environmental pollution contributes to the pathogenesis of psoriasis remain unclear. There is no significant difference between genders in the prevalence of psoriasis [23].

Psoriasis induces multidimensional health impairments, including dermatological-articular comorbidities [3], Crohn’s disease [24], nonalcoholic fatty liver disease [25], cardiovascular disease [5,26], metabolic disorders [27], and so on. Psoriasis and its comorbidities significantly impair patients’ quality of life, impose substantial economic burdens on individuals and their families, generate psychological distress, and may even threaten survival [28,29]. Furthermore, psoriasis can involve the face, hands, and reproductive organs, thereby impairing psychosocial well-being and elevating risks of depression, social withdrawal, and suicidality [30,31]. It has been reported that 43.2% of individuals living with moderate-to-severe plaque psoriasis exhibit depressive symptoms [32]. For patients with psoriasis being comorbid with depression, particular emphasis should be placed on psychological counseling and treatment.

## 3. Clinical Presentation

Based on clinical and pathophysiological characteristics, psoriasis is mainly categorized into eight subtypes, such as chronic plaque, guttate psoriasis, erythrodermic psoriasis, psoriatic arthritis (PsA), pustular psoriasis, inverse psoriasis, seborrheic psoriasis, and nail psoriasis. The clinical characteristics of psoriasis subtypes are summarized in Table 1.

Chronic plaque psoriasis, which accounts for approximately 90% of all psoriasis cases, initially presents as erythematous papules or plaques [33,34]. These lesions progressively develop into well-demarcated, polymorphic plaques adorned with thick, silvery-white scales. Upon removal of the outermost scales, lamellar scaling (referred to as the “wax-drop phenomenon”) becomes apparent. Further scraping reveals a reddish, glossy, translucent film (also known as the thin-film phenomenon). Gentle removal of this film results in punctate bleeding, known as the Auspitz sign [35]. The lesions can occur symmetrically on any part of the body, with morphological variations depending on the affected region. Both the extent and severity of cutaneous involvement closely correlate with the systemic inflammatory burden, serving as a key indicator for assessing psoriatic comorbidity risk.

Guttate psoriasis accounts for about 2% of psoriasis and predominantly affects adolescents. It typically presents with numerous scaly papules measuring 0.3 to 0.5 cm in diameter and is frequently triggered by streptococcal pharyngitis [36]. The onset of the disease is rapid, with flushing of the skin and varying degrees of itchiness. With appropriate treatment, the symptoms may ameliorate within a few weeks, and in a few patients the disease may become chronic [37].

Erythrodermic psoriasis, a severe and potentially life-threatening variant of psoriasis, accounts for approximately 1–2% of all psoriasis cases [38]. It is characterized by generalized erythema, edema, and prominent scaling, typically affecting >75% of the body surface areas [39]. Due to severe compromise of the skin barrier function, intensive therapeutic intervention is required.

PsA, an inflammatory joint disorder associated with psoriasis, may develop concurrently with or subsequent to cutaneous manifestations. Clinical features include joint swelling, pain, and progressive functional impairment. In severe cases, these may advance to irreversible joint deformities. This progressive and irreversible disease course highlights the critical importance of early diagnosis and timely intervention.

Pustular psoriasis is characterized by the eruption of sterile, superficial micropustules ranging from pinhead to millet-sized (1–3 mm). This eruption is characterized by densely aggregated pustules that may coalesce into lakes of pus. While the disease can remain confined to the palms and soles, its generalized form manifests acutely, constitutes a severe variant of psoriasis, and necessitates systemic therapy. The pustules generally desiccate and crust within 1–2 weeks, followed by spontaneous resolution. However, the disease often exhibits a relapsing-remitting course, with periodic recurrence of pustular.

Inverse psoriasis affects skin folds, such as the buttocks, groin and armpits. The skin lesions are red, shiny and without characteristic scales. Owing to the moist and friction-prone nature of the affected areas, this condition is frequently confused with or complicated by fungal or bacterial co-infections, thereby complicating both diagnosis and therapeutic management.

Seborrheic psoriasis occurs in the sebaceous areas of the scalp and face, as well as in the area behind ears or in front of the sternum. Its clinical sign exhibits the features intermediate between psoriasis and seborrheic dermatitis, necessitating differential diagnosis against both conditions.

Nail psoriasis manifests itself in a variety of ways, including small pits on the nail plate, nail separation, oil droplets (orange discoloration of the nail bed), or fragmentation of the nail plate. It affects about 40.9% of patients with plaque psoriasis, and patients with nail involvement have a longer duration of disease, higher severity of disease, and a higher frequency of PsA [40].

**Table 1 ijms-26-09233-t001:** Summary of Clinical Features of Psoriasis Subtypes.

Subtype Name	Approximate Prevalence Among Psoriasis Patients (%)	Key Clinical Features	Severity
Psoriasis vulgaris	90% [33]	Well-demarcated erythematous plaques covered with thick silvery-white scales; wax-drop phenomenon, thin-film phenomenon, Auspitz sign; symmetrically distributed, often accompanied by pruritus	Mild to moderate (most common)
Guttate psoriasis	2% [41]	Frequently occurs in adolescents, often secondary to streptococcal infection; numerous scaling papules (0.3–0.5 cm in diameter) disseminated over the body; acute onset, may resolve within weeks	Typically acute, transient, may progress to chronic phase
Erythrodermic psoriasis	1–2% [38]	Diffuse erythema and edema involving >75% of body surface area; extensive pityriasis-like scaling; accompanied by systemic symptoms such as fever and lymphadenopathy	Severe, potentially life-threatening
Psoriatic arthritis(PsA)	24% [42]	Joint swelling, pain, and limited mobility, potentially leading to deformity in severe cases; can affect large and small joints, as well as sacroiliac joints; radiographic findings include joint erosion, space narrowing, and other destructive changes	Severe, progressive
Pustular psoriasis	1–3% [43]	Sterile pinpoint to millet-sized pustules that may coalesce into “lakes of pus”; can be localized (e.g., palms and soles) or generalized	Severe
Inverse psoriasis	/	Occurs in skin folds; presents as erythematous, shiny plaques, typically without scaling	Mild to moderate
Sebopsoriasis	/	Occurs in sebaceous gland-rich areas; clinical features overlap between psoriasis and seborrheic dermatitis	Mild to moderate
Nail psoriasis	40.9% [40]	Nail plate pitting, onycholysis, “oil-drop” discoloration, and crumbling of the nail plate	Often associated with long disease duration, severe disease, and PsA

The distribution of psoriasis subtypes is heterogeneous across different countries and regions.

The diagnosis of psoriasis is primarily determined by the location, type and history of skin injury. Histopathological features have been shown to possess a certain degree of diagnostic value, while skin imaging has been identified as having auxiliary diagnostic value.

Certain diseases exhibit symptoms similar to those of psoriasis and require differential diagnosis (Table 2). For instance, atopic dermatitis typically occurs in the flexural regions of the body, such as the antecubital and popliteal fossae. The lesions lack a clear demarcation from unaffected skin. Scalp psoriasis should be differentiated from seborrheic dermatitis and tinea capitis. Psoriatic lesions on the trunk and extremities must be distinguished from secondary syphilitic eruptions, lichen planus, and chronic eczema. PsA requires differentiation from rheumatoid arthritis (RA) in diagnosis. Inverse psoriasis should be differentiated from candidiasis and other fungal infections. Pityriasis rosea can be distinguished from guttate psoriasis by its fine scaling and the presence of a herald patch, which appears 1–2 weeks before the generalized eruption. Additionally, early lesions of cutaneous T-cell lymphoma lack well-defined borders and thick scales, unlike those of psoriasis.

## 4. Pathophysiology

### 4.1. Histological Features

The histopathological characteristics of psoriasis include epidermal hyperplasia, hyperkeratosis with parakeratosis, presence of Munro microabscesses (focal collections of neutrophils within the epidermis), and marked diminution or complete absence of the granular layer [23]. Psoriasis shows twisted and widened capillaries in the dermal papillae with swollen lining cells and thinning of the suprapapillary epidermal plates. A mixed inflammatory infiltrate composed predominantly of T cells with some neutrophils is observed in both the dermis and perivascular areas. Erythrodermic psoriasis is characterized by marked dilation and congestion of superficial dermal vasculature, while pustular psoriasis exhibits the pathognomonic Kogoj micropustules [44].

### 4.2. Genetic Contributions

Psoriasis demonstrates strong genetic predisposition. Genome-wide association studies (GWAS) have identified 109 distinct susceptibility loci for psoriasis [45,46,47]. Approximately 30% of psoriasis patients report a family history, and the disease concordance rate in monozygotic twins reaches 68% [48], highlighting the significant genetic contribution to psoriasis pathogenesis. Environmental triggers may modulate disease phenotype through immunoregulatory and epigenetic mechanisms [49,50]. The interplay between genetic and environmental factors leading to immune system activation is illustrated in Figure 1. *HLA-C**06, a human leukocyte antigen (HLA) allele, triggers psoriasis by inducing an autoimmune response against melanocytes through presentation of the “a disintegrin and metalloproteinase with thrombospondin motifs-like 5” (ADAMTSL5) autoantigen [51]. The *erap1* gene encodes an interferon (IFN)-γ-induced aminopeptidase that in the endoplasmic reticulum trims precursor peptides of melanocyte autoantigens to the right length for *HLA-C**06:02 presentation, activating CD8^+^ T-cell-mediated autoimmune disease [52]. In addition to the genes previously mentioned, *HLA-DRB1**07 [53], *il23r* [54], *ddx58* [55], *traf3ip2* [56] and other genes have also been found to be associated with psoriasis [57].

### 4.3. Triggers

In addition to genetic predisposition, environmental factors, hormonal levels, and lifestyles also contribute to the onset and progression of psoriasis. Microbial infections have been strongly implicated in both disease initiation and exacerbation. For example, infections with *Streptococcus pyogenes* may trigger or exacerbate psoriasis through superantigen-mediated mechanisms or epitope spreading [58,59]. CD8^+^ T cells in psoriasis patients cross-recognize streptococcal M proteins and keratin 17 (K17) via molecular mimicry [60,61]. Psoriasis incidence increased in HIV-infected individuals compared to the general population [62]. HIV may directly trigger psoriasis as a source of superantigens or as a co-stimulatory factor for antigen delivery [63], and activated CD8^+^ T cells produce more IFN-γ during HIV infection [64], or as HIV infection destroys CD4^+^ T cells, thereby affecting the immunomodulatory role of regulatory T cells (Tregs). In addition to the aforementioned pathogens, intestinal dysbiosis has been consistently associated with increased psoriasis incidence [65,66,67]. 31–88% of psoriasis patients experienced stressful events prior to disease onset, and individuals experiencing significant stress within 12 months demonstrate higher psoriasis incidence [68]. It was proposed that stress may mediate the pathophysiology of psoriasis through the hypothalamic–pituitary–adrenal axis (HPA), the immune pathway and the peripheral nervous system [69].

Fluctuations in estrogen levels can potentially impact the progression of psoriasis. Estrogen may exert anti-psoriatic effects via estrogen receptors α and β (ERα/ERβ)-mediated downregulation of proinflammatory cytokines in immune cells. Paradoxically, under specific conditions, estrogen can promote IL-23 secretion by dendritic cells, potentially exacerbating disease [70,71,72], suggesting that estrogen may play dual regulatory roles on psoriasis in an environmentally dependent manner [71,73]. Lifestyle also influences the course of psoriasis. Emerging evidence indicates that elevated cortisol levels in sleep disorders such as insomnia stimulate cutaneous mast cells, impair skin barrier function and upregulate proinflammatory cytokine expression, thereby exacerbating psoriasis [74,75]. Furthermore, research has indicated that individuals with a history of cigarette smoking exhibit a higher incidence of psoriasis compared to the general population [47]. At present, there are still no sufficient evidence to determine whether alcohol consumption is associated with the development and recurrence of psoriasis [76,77,78,79]. Thus, efforts will be required to investigate the effect of alcohol on psoriasis.

### 4.4. Pathogenesis

The pathogenesis of psoriasis involves intricate interactions of genetic, immunological, and environmental factors, characterized by the synergistic dysregulation of immune system activation and epidermal cell dysfunction [80,81]. This process is driven by aberrant immune responses, which fuels a self-amplifying cascade of IL-17 and IL-23 secretion, resulting in the hallmark pathological features of chronic inflammation (erythema) and epidermal hyperproliferation (scaling). Cutaneous injury or microbial infection leads to the release of damage-associated molecular patterns (DAMPs) from host cells and pathogen-associated molecular patterns (PAMPs) from pathogens. Together, these molecules activate Toll-like receptor (TLR) signaling in various skin cells, triggering a proinflammatory cytokine cascade that drives the chronic inflammation seen in psoriasis [82].

Keratinocytes (KCs), constituting approximately 90% of epidermal cells, exhibit abnormal proliferation and differentiation that directly contribute to psoriatic acanthosis [83,84]. Beyond DAMPs activation, KCs respond to both exogenous antigens and endogenous autoantigens by producing antimicrobial peptides, S100 proteins, and other effector molecules that initiate or amplify innate and adaptive immune responses, thereby participating in psoriatic inflammation [55,85]. Research demonstrates that KC-derived S100A9 induces dendritic cells to produce IL-23, driving the IL-23/IL-17 axis, while IL-17A further stimulates KCs to express IL-25 (also termed IL-17E), which mediates proinflammatory phenotypes and hyperproliferation through signal transducer and activator of transcription (STAT) 3 activation [86]. Additionally, KC-secreted chemokine C-C motif chemokine ligand 20 (CCL20) recruits CD4^+^ T helper (Th)-17 cells and γδ T cells, exacerbating inflammatory responses [87]. KCs also respond to cytokines such as IFN-γ, IL-17, and IL-36, establishing positive feedback loops [88,89,90]. Notably, loss of protein phosphatase 6 in keratinocytes activates the C/EBP-β-ARG1 pathway, increasing polyamine (e.g., spermidine) production. These polyamines bind self-RNA, enhancing dendritic cell recognition through TLR7 signaling and worsening psoriasis inflammation [91].

### 4.5. Innate Immunity

Aberrant activation of the innate immune system may contribute to psoriatic inflammation, wherein immune cells including neutrophils, dendritic cells, and macrophages become hyperactivated through pattern recognition receptor (PRR)-mediated detection of DAMPs or PAMPs, subsequently releasing excessive proinflammatory cytokines that further stimulate adaptive immune responses (as shown in Figure 2).

In psoriatic lesions, neutrophils accumulate within the epidermal compartment [92,93]. Alarmins, selectins, and cytokines mediate neutrophil recruitment to inflammatory sites [94,95]. The recruited neutrophils release neutrophil extracellular traps (NETs) [96,97]. These NETs induce IL-36γ and lipocalin-2 release from keratinocytes via the TLR4/IL-36R axis, thereby promoting further neutrophil recruitment and sustaining inflammation [98,99]. Furthermore, the neutrophils promote the initiation of inflammatory responses and early lesion development through interactions with macrophages, DCs, and other immune cells [100]. Neutrophil-derived IL-17A potently induces KCs to express proinflammatory cytokines and chemokines, enhancing their migratory phenotype and infiltration [101,102]. Concurrently, migrating neutrophils overexpress matrix metalloproteinase-9 (MMP-9), promoting psoriasis progression by inducing vascular remodeling, enhancing endothelial activation, and driving the accumulation of CD4^+^ T cells [103]. Interestingly, it was reported that roflumilast-loaded Ly6-targeted immunonanocarriers, which specifically bind to neutrophils, significantly alleviated psoriasiform dermatitis by suppressing neutrophil activation and inflammatory responses [104].

In healthy skin tissue, dendritic cells (DCs) are predominantly confined to the dermal compartment, whereas approximately 50% of DCs migrate to the epidermal layer in psoriatic lesions [105,106,107]. Notably, the recruited DCs in psoriatic lesions stimulate a distinct subset of dual-function T cells that secrete IFN-γ and IL-17, a phenomenon not observed in non-lesional areas or healthy skin [108]. The IFN-γ exacerbates skin barrier dysfunction by inducing apoptosis in KCs [109], while IL-17 promotes hyperproliferation of KCs [89], collectively resulting in characteristic psoriatic acanthosis and chronic inflammation. Plasmacytoid DCs (pDCs) infiltrating psoriatic lesions induce IFN-α secretion, driving T-cell expansion. Xenotransplantation experiments further demonstrated that blocking IFN-α signaling completely prevents psoriasis development [110].

Cutaneous macrophages are broadly categorized into epidermal Langerhans cells (LCs) and dermal macrophages. As tissue-resident immune sentinels, LCs are critical in the early pathogenesis of psoriasis [111], where activated keratinocytes in psoriatic lesions overproduce bone morphogenetic protein 7 (BMP7) to promote progenitor differentiation into proinflammatory LCs [112]. These LCs secrete proinflammatory cytokines, driving the accumulation of CD4^+^ T cells and γδ T cells in psoriatic skin [113,114]. Furthermore, psoriatic LCs demonstrate upregulated expression of S100A8/S100A9 proteins that stimulate IL-15 production, while concurrently generating IL-1β and IL-23 [115]. Notably, LCs in psoriasis patients exhibit upregulation of C-X-C motif chemokine ligand 9 (CXCL9), which mediate the targeted recruitment of T lymphocytes to inflammatory foci, thereby amplifying the psoriatic immune cascade [116].

Psoriatic lesions demonstrate increased activated macrophages, particularly during late-stage inflammation [106,117]. Patients with psoriasis show elevated circulating monocyte-derived macrophages in plasma that predominantly exhibit an M1 phenotype; these monocytes infiltrate the dermo-epidermal junction of skin lesions, serving as critical sources of inflammation [118,119]. In psoriasis, IL-17 directly promotes inflammation and activates proinflammatory polarization of monocyte-macrophages, while also enhancing their sensitivity to pathogen-derived TLR4 ligands, thereby amplifying inflammatory signaling [120]. It was found that in the CD18 hypomorphic psoriasiform mouse model, activated macrophages played a pivotal role in chronic inflammation through TNF-α release, with either macrophage depletion or TNF-α neutralization significantly ameliorated skin pathology [118]. Furthermore, macrophages contribute to psoriatic pathogenesis not only through secretion of proinflammatory cytokines but also via intricate interactions with T lymphocytes, dendritic cells, and other immune cells, collectively driving disease progression [121].

Mast cells are critical in acute allergic and chronic inflammatory pathologies [122,123]. Upon IFN-α stimulation, mast cells release exosomes that translocate cytosolic PLA2 enzymatic activity to juxtaposed CD1a-expressing antigen-presenting cells, thereby instigating a cytokine milieu dominated by IL-22 and IL-17A secretion [124]. It was reported that the suprabasin (SBSN)-derived polypeptide, SBSN (50-63), activates mast cells via TLR4 signaling, exacerbating inflammatory responses in psoriasis [125]. Furthermore, the Koebner phenomenon in psoriasis is likely mediated by trauma-induced mast cell activation and subsequent release of inflammatory mediators [126].

γδ T cells significantly contribute to psoriatic inflammation, with Vγ5^+^γδ T andVγ4^+^γδ T cell subsets exhibiting specific cutaneous tropism [127]. It has been demonstrated that IL-1β, acting through the IL-1R axis, drives γδ T-cell activation and expansion [128]. Proinflammatory cytokines secreted by γδ T cells activate keratinocytes, which in turn produce inflammatory mediators that collectively promote inflammatory responses in psoriasis [129].

Natural killer cells (NKs) play a significant immunomodulatory role in psoriasis through cytokine-mediated regulation of Th1/Th2 and Th17/Treg cell network, thereby contributing to the disease pathogenesis [130,131,132]. Research has revealed that psoriasis patients exhibit an increased proportion of the IL-17-secreting NKT cell subset, and targeted therapy against IL-17 can effectively correct this imbalance [133].

Myeloid-derived suppressor cells (MDSCs) typically secrete immunosuppressive cytokines like IL-10 and TGF-β to downregulate the expression of proinflammatory cytokines and inhibit immune responses. However, emerging evidence points to a paradoxical role for these cells in psoriasis. It was found that MDSCs from psoriasis patients paradoxically promoted Th17 responses by enhancing Th17 cell differentiation, increasing production of IL-23, IL-1β, and CCL4, while showing reduced expression of PD-1 and PD-L1 and failing to generate Tregs, thereby disrupting T cell homeostasis [134]. Furthermore, in psoriasis, myeloid-derived adjuster cells (MDACs) drive pathological inflammation by promoting differentiation into proinflammatory M1 macrophages and DCs. Notably, the recent literature indicates that suppression of the RORγt/NFAT1 axis can restore the ability of MDACs to suppress T cell activity in psoriasis [135].

### 4.6. Adaptive Immunity

It was found that psoriatic skin lesions are enriched with CD4^+^ T cells [121]. Upon skin injury or infection, DAMPs are released and activate DCs via TLR7/8, triggering the release of proinflammatory cytokines, notably TNF-α and IL-17, which drive the differentiation of CD4^+^ T cells into Th17 and Th1 subsets. Th17 cells produce cytokines including IL-17A, IL-21 and IL-22, with IL-17A directly activating the NF-κB and MAPK cascades in KCs, inducing the expression of antimicrobial peptides (AMPs) and proinflammatory factors, thereby promoting epidermal hyperplasia and amplifying inflammation [136,137,138,139]. IL-21 and IL-22 disrupt the epithelial barrier, increasing epithelial exposure to pathogens or DAMPs, which perpetuates DC activation [140,141]. Furthermore, under the influence of IL-17 and IL-6, DCs secrete elevated levels of IL-23, a cytokine essential for the development and functional maturation of Th17 cells, thereby further expanding the Th17 population [142].

Th1 cells produce IFN-γ and TNF-α, where IFN-γ enhances the antigen-presenting capacity of DCs and promotes M1 macrophage polarization (secreting IL-12 and NO), thereby reinforcing Th1 differentiation. TNF-α directly contributes to tissue damage, releasing additional DAMPs that sustain TLR7/8 pathway activation, establishing a proinflammatory positive feedback loop [121,143].

The Th22 cell is characterized by producing IL-22, which promotes abnormal KCs proliferation and parakeratosis while establishing a positive feedback loop through CCL20 induction [144,145]. Th9 cells display skin tropism and tissue-resident properties in cutaneous environments. Notably, psoriatic lesions exhibit significantly elevated levels of both Th9 cells and their secreted IL-9 compared to healthy skin controls [146]. Mechanistically, IL-9 contributes to psoriasis pathogenesis by potentiating Th17-mediated inflammatory responses and promoting pathological angiogenesis [147]. The frequency of T follicular helper (Tfh) cells increased in both peripheral blood and lesional skin of psoriasis patients, and this increase correlated positively with IL-21 levels and disease severity [148].

However, in psoriasis patients, Th2 cells and their secreted cytokines, IL-4 and IL-10, are reduced [149]. Studies have shown that IL-4 suppresses IL-23 secretion by DCs, thereby limiting Th17 activation [150]. Additionally, activation of acid-sensing ion channel 3 (ASIC3) triggers the production of calcitonin gene-related peptide (CGRP) from sensory neurons, subsequently enhancing IL-23 production by DCs and thereby amplifying Th17 cell activation [151]. These findings not only delineate the systemic inflammatory characteristics in psoriasis but also highlight the pivotal role of the IL-23/IL-17 signaling axis in driving cutaneous inflammation.

It was reported that specific K17 truncated variants (aa118-132) exhibited *HLA-DRB1*-restricted immunogenicity, inducing T cell proliferation and IFN-γ production in psoriatic patients in vitro [53]. K17 may also trigger CD8^+^ T cell-specific immune responses through an *HLA-C**06:02-restricted mechanism, with this differential MHC restriction likely attributable to its multiple antigenic epitopes. It was found that the presence of peptidoglycan (PG)-specific Th1 cells in psoriatic lesions that recognize streptococcal or staphylococcal PG in an *HLA-DR*-restricted manner and secrete IFN-γ, with enhanced PG recognition exacerbating *streptococcus*-induced psoriasis [152,153]. Furthermore, the presence of autoantigens (e.g., ADAMTSL5, PLA2G4D) and pathogenic immune complexes (e.g., polyamine-RNA-peptide, LL37-self DNA/RNA), and the consequent activation of immune responses, underscore the pivotal role of T cell-mediated immunity in disease pathogenesis. Psoriatic lesions exhibit predominant expression of T cell receptor (TCR) Vβ2 and Vβ6 gene families, with CDR3 sequence analysis revealing oligoclonal expansion, supporting an antigen-driven T-cell response [154]. In *HLA-C**06:02-positive individuals, specific CD8^+^ T cells expressing Vα3S1/Vβ13S1 TCRs demonstrate cytotoxic activity against *HLA-C**06:02-expressing melanocytes, confirming antigen-specific T cell activation through HLA-mediated presentation of autoantigens such as ADAMTSL5 [51]. However, subsequent studies analyzing the TCR repertoire in five psoriasis patients revealed heterogeneous Vβ gene usage patterns without consistent expansion of Vβ6.1-3 or Vβ13.1 clones, suggesting potential interpatient variability in T cell responses [155].

CD8^+^ T cells exhibit significant infiltration and activation in psoriatic lesions, particularly within the epidermal layer. These cells are activated through antigen presentation by DCs and the cytokine microenvironment, with IL-23 promoting their differentiation into Tc17 cells. Tc17 cells secrete proinflammatory cytokines, directly driving epidermal hyperplasia and inflammatory responses [156]. It was found that IFN-γ produced by Tc17 cells activated the Janus Kinase (JAK) 1/STAT1 signaling pathway, upregulating KCs expression of K17 and exacerbating aberrant epidermal proliferation. Meanwhile, Tc22-derived IL-22 directly stimulated KCs proliferation while inhibiting differentiation, leading to the characteristic psoriatic hyperkeratosis [157]. Furthermore, CD8^+^ T cells interacted with DCs to promote CD4^+^ T cell differentiation into Th1, Th17, and Th22 subsets, influencing KCs apoptosis, and with fibroblasts to induce CCL20 secretion, enhancing inflammatory cell tissue residency [158]. Additionally, CD8^+^ T cells recognized multiple psoriasis-associated autoantigens via MHC class I molecules. For instance, ADAMTSL5, produced by melanocytes, was presented by *HLA-C**06:02, activating CD8^+^ T cells and inducing IL-17 and IFN-γ secretion [51]. Therapeutic strategies depleting CD8^+^ T cells have shown significant efficacy [159], further underscoring their pivotal role in psoriasis pathogenesis.

Compared to healthy skin, psoriatic lesions exhibit a 100-fold increase in T cell numbers, and a 50-fold increase compared to non-lesional skin, indicating significant tissue-resident memory T cells (TRM) expansion in psoriatic plaques [160]. The recurrence of psoriasis at previously affected sites can be attributed to the accumulation and reactivation of TRM cells at these locations [161,162]. Skin TRM cells exhibit remarkable longevity, persisting for more than a year even without local antigen presentation in murine models [160,163]. The development of therapeutic agents targeting aberrantly activated T cells in psoriatic skin, particularly through modulation of their immunological memory functions to prevent disease recurrence, may hold significant potential in finding an alleviation or complete cure for psoriasis.

The functional and phenotypic alterations of Tregs in psoriasis remain incompletely understood. It was found that although the number of Treg increased systemically in psoriasis patients, their migration and survival within psoriatic lesions were impaired, resulting in insufficient immunosuppressive capacity and uncontrolled inflammation [164]. Tregs from psoriatic lesions and peripheral blood exhibit diminished capacity to suppress effector T cell proliferation [165,166]. With compromised immunosuppressive function, psoriatic Tregs paradoxically acquire the ability to drive inflammation through IL-17 secretion [166]. Furthermore, obesity-related factors, particularly long-chain fatty acids, have been shown to reduce cutaneous Treg populations, thereby aggravating psoriatic pathogenesis [167].

Psoriasis patients exhibit increased levels of B cell activation in peripheral blood, correlating with the severity of clinical symptoms. It was reported that differential proportions of CD19^+^ B cell subsets in peripheral blood among various psoriasis subtypes were observed, with these alterations closely associated with clinical disease activity, particularly during flare-ups [168]. Notably, autoantibodies against K13 [169], heterogeneous nuclear ribonucleoprotein A1 (hnRNPA1) [170], K17 [60,171], and Rab coupling protein isoform 3 (FLJ00294) [169] have been identified in psoriasis patients. Additionally, IgG autoantibodies targeting LL-37 and ADAMTS-L5 were observed in a subset of psoriasis patients [172]. Surprisingly, a 2024 case report documented complete and sustained remission of psoriasis following CD19 Chimeric antigen receptor T-cell (CAR-T) cell therapy targeting B cells, with no relapse observed over 3.5 years of follow-up [173]. Although the presence and role of B cells in psoriatic lesions remain elusive, current findings suggest their involvement in disease pathogenesis through multiple mechanisms, including autoantibody production and potential antigen presentation.

Regulatory B cells (Bregs) suppress psoriasiform inflammation through IL-10-dependent mechanisms, which concurrently enhance Tregs expansion and restrain Th17 differentiation [174]. However, it was found that a reduction in Bregs in both peripheral blood and psoriatic lesions showed an inverse correlation with increased populations of IL-17- and IFN-γ- producing T cells [175,176]. Future investigations into therapeutic strategies aimed at enhancing Bregs functionality for optimized psoriasis management should be highly encouraged.

## 5. Management and Treatment

### 5.1. Conventional Topical Therapy

Current treatment strategies for psoriasis primarily involve immunosuppression, anti-inflammatory therapy, and palliative care (Conventional treatment approaches are shown in Table 3). In addition to cutaneous and articular manifestations, psoriasis is associated with various comorbidities that require prompt therapeutic intervention.

Corticosteroids serve as first-line topical therapeutic agents, demonstrating favorable safety and tolerability profiles. Clinically, formulations of varying potencies and durations of action can be selected based on disease severity, enabling rapid symptom control [177,178]. Glucocorticoids exert multiple functions through the leucine zipper protein (GILZ), which suppresses Th17-mediated inflammation by regulating Th17-inducing cytokine expression in dendritic cells and inhibiting Th17 cell proliferation and phenotypic expression in T cells [179]. However, prolonged use of topical corticosteroids may induce localized adverse cutaneous reactions, including tachyphylaxis and suppression of the HPA axis [180].

Salicylic acid-based drugs are utilized in the treatment of psoriasis, as they promote desquamation of hyperkeratotic scales and enhance the penetration of concomitant topical therapies, such as corticosteroids [181]. However, their efficacy as monotherapy is limited. Caution is warranted regarding potential systemic absorption, particularly when applied to extensive body surface areas or used concurrently with other topical medications.

Calcipotriol, a vitamin D3 analog, primarily acts by binding to vitamin D receptors on keratinocytes to modulate keratinocyte proliferation and differentiation. It is commonly used for topical treatment of plaque psoriasis [182]. Clinical trials have demonstrated that calcipotriol exhibits comparable efficacy to most topical corticosteroids in the treatment of mild plaque psoriasis, along with a favorable safety profile. Combination therapy with vitamin D3 analogs and topical corticosteroids significantly reduces the incidence of adverse events [182,183].

Tapinarof cream, an aryl hydrocarbon receptor modulator, regulates IL-17 expression and skin barrier proteins (such as filaggrin), demonstrating significant efficacy in mild-to-severe plaque psoriasis while potentially causing local adverse reactions including folliculitis [184].

**Table 3 ijms-26-09233-t003:** Conventional Drug Therapies for Psoriasis.

Category	Drug Name	Target/ Mechanism	Efficacy Profile	Adverse Effects /Potential Risks	Notes
Topical Therapies	Corticosteroids [178]	Suppression of Th17-mediated inflammation; modulation of dendritic and T-cell activity	Rapid symptom control; suitable for mild-to-moderate plaque psoriasis	hypothalamic–pituitary–adrenal axis suppression with prolonged use	Avoid continuous long-term application
	Calcipotriol (Vitamin D3 analog) [185]	Regulation of keratinocyte proliferation/differentiation; immunomodulation	Improves hyperproliferation and differentiation; safe and effective	Local irritation (erythema, pruritus)	First-line for plaque psoriasis
	Tacrolimus ointment [186]	Calcineurin inhibition	High local tolerability	Burning sensation, infection risk	Suitable for mild cases or combination therapy
	Calcipotriol/betamethasone combo [187]	Synergistic anti-inflammatory effects of Vitamin D3 analog combined with corticosteroid	Superior efficacy to monotherapy; reduced relapse	Reduced skin irritation vs. monotherapy	Indicated for moderate-to-thick plaques
	Tapinarof cream [184]	Aryl hydrocarbon receptor, modulation; IL-17 suppression; skin barrier enhancement	Significant improvement in mild-to-severe plaque psoriasis	Folliculitis, local irritation	Favorable safety profile
Systemic Therapies	NB-UVB [188]	local immunosuppression	First-line therapy with high safety	Frequent clinic visits; limited phototherapy center access	Suitable for pregnant patients and children
	MTX [178]	Folate metabolism inhibition; anti-inflammatory	Effective for moderate-to-severe psoriasis and psoriatic arthritis	Hepatotoxicity myelosuppression, gastrointestinal disturbances	Requires regular liver function monitoring
	Cyclosporine [189]	Selective T-cell inhibition; IL-2 blockade	Rapid onset; used for acute flares	Nephrotoxicity, immunosuppression	renal monitoring
	Acitretin [190,191]	Keratinocyte differentiation modulation; anti-inflammatory	Controls hyperkeratosis; adjunct for severe psoriasis	Teratogenicity, xerosis, photosensitivity	Prohibited for pregnant women

Due to space limitation, only representative drugs per category are listed.

### 5.2. Conventional Systemic Therapy

Phototherapy is considered a first-line intervention for moderate-to-severe psoriasis following the failure of topical therapies. Ultraviolet B (UVB) radiation penetrates the epidermis directly, inducing the formation of pyrimidine dimers in DNA, which subsequently triggers apoptosis of activated and proliferating T cells [192,193]. Narrowband UVB (NB-UVB) downregulates the expression of multiple proinflammatory cytokines and promotes the release of anti-inflammatory mediators, such as interleukin-10 (IL-10), thereby reversing the inflammatory milieu in psoriatic lesions [194]. Evidence-based medicine has demonstrated that NB-UVB phototherapy exhibits well-established efficacy and an excellent safety profile, making it suitable for nearly all patient populations, including children and pregnant women [188]. Despite its efficacy, clinical implementation is limited by the scarcity of phototherapy centers and need for frequent hospital visits, creating practical burdens for patients.

Acitretin, a synthetic retinoid used for moderate-to-severe psoriasis, has been shown to enhance therapeutic efficacy when combined with other systemic treatments while allowing dose reduction and minimizing adverse effects [195,196,197]. Acitretin primarily exerts its therapeutic effects by activating retinoic acid receptors (RARs), thereby modulating the expression of a series of target genes [198]. This leads to the inhibition of abnormal keratinocyte proliferation and the promotion of their normal differentiation [199]. However, robust clinical trial data supporting its efficacy and safety as monotherapy remain insufficient. Of particular importance is its well-documented teratogenicity, which mandates absolute contraindication in women of childbearing potential and continuation of effective contraception for at least three years following treatment [191,200].

Methotrexate (MTX), a folate antagonist, is recommended for the systemic treatment of moderate-to-severe psoriasis and psoriatic arthritis because it interferes with purine and pyrimidine synthesis by inhibiting dihydrofolate reductase (DHFR), thereby exerting antiproliferative and immunomodulatory effects. Serious adverse effects include hepatotoxicity and immunosuppression [201].

Cyclosporine, a calcineurin inhibitor, is indicated for moderate-to-severe psoriasis and PsA [189,202]. Cyclosporine potently and selectively inhibits the activation and proliferation of T cells [203], thereby effectively suppressing the subsequent cytokine storm [204]. This mechanism fundamentally disrupts the core immunological drivers of psoriatic inflammation. This drug demonstrates rapid therapeutic onset and is associated with a relatively low risk of myelosuppression and hepatotoxicity. Major adverse effects include nephrotoxicity, immunosuppression, and potential malignancy [205].

### 5.3. Biologicals

The ideal therapeutic approach should precisely target pathological immune responses while preserving protective immunity, though this goal requires further investigation to achieve. In cases where conventional systemic therapies demonstrate inadequate efficacy, significant adverse effects, or contraindications due to comorbidities, biologicals have emerged as the treatment of choice.

Antibodies: Monoclonal antibody demonstrates significant efficacy in psoriasis treatment, and bispecific antibodies targeting dual or multiple signaling pathways exhibit synergistic therapeutic effects. Ustekinumab (anti-IL-12/23p40 antibody) [206] and Tildrakizumab (anti-IL-23p19 antibody) [207] have been successfully employed in psoriasis management. The anti-IL-17A antibodies Ixekizumab and Secukinumab demonstrate both safety and clinical efficacy in psoriasis treatment [208,209]. Clinical trials indicate the bispecific antibody Bimekizumab (targeting IL-17A and IL-17F) shows superior efficacy in alleviating psoriasis symptoms compared to inhibitors targeting either IL-17 or TNF-α alone, though with higher incidence of adverse effects including oral candidiasis and diarrhea [210,211]. Spesolimab, an IL-36 receptor monoclonal antibody, effectively controls acute flares and reduces recurrence risk in generalized pustular psoriasis [212]. Cyclophilin A (CypA), a proinflammatory factor interacting with ACE2 and CD147 to promote psoriatic inflammation, shows promising therapeutic potential as anti-CypA monoclonal antibody demonstrates superior efficacy compared to combined anti-IL-17A antibody and methotrexate therapy [213].

Small RNA-mediated gene silencing strategy: Small interfering RNAs (siRNAs) have demonstrated therapeutic potential for psoriasis in recent years. MiR-340 specifically binds to the 3′UTR in mice, targeting and suppressing IL-17A expression, thereby alleviating disease severity in imiquimod-induced psoriatic mouse models [214]. Additionally, intradermal injection of HMGCS1 siRNA modulates immune responses and keratinocyte function, reducing PASI scores, epidermal hyperplasia, and IL-23 expression while inhibiting STAT3 phosphorylation levels in psoriatic mouse models [215]. The rapid advancement of siRNA applications is closely associated with progress in lipid nanoparticle (LNP) technology and nucleic acid modification methods [216,217]. Utilizing a lipid-based nanocarrier (CYnLIP) to co-deliver IL-36α siRNA and erlotinib (a classical tyrosine kinase inhibitor) significantly decreased psoriasis-like plaque severity (PASI score reduction from 4 to 1) and improved pathological features by suppressing inflammatory pathways in murine models [218]. These findings indicate that siRNA and miRNA-based technologies, through targeted gene suppression, offer novel strategies and approaches for psoriasis treatment.

CAR-T immunotherapy: CAR-T is an innovative therapeutic approach that combines molecular biology, virology, and immunology to genetically engineer T-cells with synthetic CARs, enabling MHC-independent target cell recognition and elimination for precision medicine. While achieving groundbreaking success in treatment of cancers, its precise targeting capability has expanded applications to autoimmune diseases [1]. In 2024, it was reported that the complete remission of a 45-year chronic plaque psoriasis coincided with CD19 CAR-T treatment for relapsed large B-cell lymphoma [173]. Subsequently in 2025, CD19 CAR-T therapy induced profound remission in precursor B-cell acute lymphoblastic leukemia while simultaneously resolving refractory severe plaque psoriasis [219]. CD19 CAR-T therapy unexpectedly ameliorated psoriatic symptoms through B-cell depletion. These findings suggest that CD19-targeted CAR-T cells may restore immune homeostasis and suppress autoimmune responses in psoriasis by eliminating aberrantly activated B cells or modulating T-cell subsets. Furthermore, it was found that CRISPR-engineered allogeneic CD19 CAR-T cells safely induced durable remission in refractory autoimmune diseases, including severe myositis and systemic sclerosis, by achieving profound B cell depletion and reversing tissue damage [220]. This groundbreaking study laid a crucial foundation for the clinical advancement of universal CAR-T cell therapy.

Treg-targeted strategy: Tregs are crucial for maintaining immune tolerance by suppressing effector T cell response, and restoring Treg number and function has become a novel therapeutic target in psoriasis treatment. Genetically engineered CAR-FoxP3 Tregs, administered intranasally, effectively suppressed inflammation, ameliorated disease symptoms, and conferred sustained protection against secondary experimental autoimmune encephalomyelitis (EAE) induction in a murine model of EAE [221]. CAR-Tregs represent a novel therapeutic strategy for autoimmune diseases by “restoring immune tolerance”, offering promising applications in the treatment of psoriasis. It was reported that a perforated microneedle (PMN) system loaded with Tregs significantly improved inflammatory symptoms in a psoriasis mouse model by locally delivering cells and releasing propionic acid to enhance Treg function [222]. Furthermore, a clinical trial demonstrated that low-dose IL-2 can safely and selectively expand Tregs, showing potential clinical efficacy in various autoimmune diseases, including psoriasis [223]. However, the broad application of Treg-targeted strategies remains constrained by key challenges, including the complexity of bioengineering techniques, optimization of cell delivery efficiency and tissue-specific targeting, long-term stability of therapeutic effects, and mitigation of risks linked to non-specific immunosuppression. Therefore, future research must focus on addressing these bottlenecks to achieve safe, durable, and personalized immune tolerance induction.

Hematopoietic stem cell transplantation (HSCT): HSCT employs myeloablative conditioning to eliminate pathogenic immune cells (including T and B lymphocytes) and reconstitute normal hematopoietic and immune systems. This therapy is primarily indicated for patients with treatment-refractory severe psoriasis, particularly those with comorbid autoimmune disorders or hematologic abnormalities. Case reports have documented complete remission of both immunoglobulin light-chain amyloidosis and psoriasis following autologous HSCT [224,225]. Due to significant risks including infections, graft-versus-host disease (GVHD), secondary malignancies, and other autoimmune conditions, HSCT is strictly limited to carefully selected severe cases after thorough risk-benefit evaluation. Future research should focus on optimizing regimens, refining patient selection criteria, and developing strategies to reduce complications through clinical trials.

Mesenchymal stromal cells (MSCs) are adult stromal cells possessing self-renewal capacity, multilineage differentiation potential, and immunomodulatory properties. Their low immunogenicity enables allogeneic transplantation. Clinical trials demonstrate that psoriatic patients receiving allogeneic umbilical cord-derived MSCs (UMSCs) transplantation exhibited favorable safety profiles with significant therapeutic effects observed in some cases [226,227]. In 2024, it was reported that intravenous administration of adipose-derived MSCs (AD-MSCs) in moderate-to-severe psoriasis showed good safety and tolerability [228]. Furthermore, some patients achieved long-term remission [229], providing a foundation for larger-scale investigations. However, current evidence primarily derived from small-scale trials or short-term follow-up, requires further validation of long-term risks and treatment protocol standardization. Future directions should include multicenter randomized double-blind controlled trials to provide high-level evidence, reduce production costs, and improve treatment accessibility. The paracrine mechanism represents the primary therapeutic mode of MSCs, involving both direct cell–cell contact and secretion of soluble factors to modulate immune responses. In psoriasis treatment, significant clinical effects can be achieved using only MSC-conditioned medium (MSC-CM) without direct MSC administration [230]. This strategy avoids potential risks associated with live cell transplantation while offering better standardization potential and improving clinical translation prospects.

Exosomes are 30–150 nm vesicular particles capable of transporting proteins, nucleic acids (e.g., miRNA), and lipids, participating in immunomodulation and inflammatory responses. In psoriasis, MSC-derived exosomes restore immune balance by delivering molecules such as miR-146a to inhibit Th17 differentiation and promote Treg proliferation [231]. Clinical studies demonstrate that adipose-derived mesenchymal stem cell exosomes exhibit optimal therapeutic efficacy, significantly improving erythema, induration, and scaling while modulating immune homeostasis [232]. Furthermore, dimethyl fumarate (DMF)-loaded Treg exosomes (rExo@DMF MNs) delivered via microneedles reduce proinflammatory cytokine release, inhibit keratinocyte proliferation and migration, and ameliorate psoriasiform inflammation in murine models [233]. Future directions may involve exosome-based drug delivery systems (e.g., microneedles, ointments) as novel therapeutic approaches for psoriasis [233,234]. However, exosome therapy faces challenges including heterogeneity and difficulties in clinical-scale production.

Bispecific T-cell engagers (BiTEs) represent a therapeutic approach that redirects T-cell cytotoxicity against specific target cells by simultaneously engaging CD3 on T cells and disease-associated antigens. The CD19xCD3 BiTE blinatumomab mediates T-B cell conjugation to induce B-cell apoptosis, demonstrating clinical improvement in rheumatoid arthritis patients even at low doses [235], while no clinical data yet exists for psoriasis.

Gene therapy: Targeting genetically altered loci or their transcriptional products in psoriatic patients represents a promising therapeutic strategy. As discussed earlier, the modulation of specific gene transcripts using siRNA approaches will not be further elaborated. It was demonstrated that CRISPR/Cas9-mediated PD-L1 overexpression ameliorated psoriatic manifestations in murine models [236]. Furthermore, the application of CRISPR/Cas9 to target NLRP3 has been shown to ameliorate the inflammatory response in murine models [237]. Current research on CRISPR/Cas9-based therapies for psoriasis remains exploratory due to limitations in delivery systems and off-target effects. It was found that the elevated level of H4K16 acetylation (H4K16ac) in macrophages from psoriatic lesions showed positive correlation with self-RNA accumulation and disease severity, the lysine acetyltransferase8 (KAT8) was recruited by AP-1 transcription complexes to CXCL2/CCL3 chemokine promoter regions, promoting neutrophil infiltration and NETs formation, and that genetic ablation or pharmacological inhibition of KAT8 substantially attenuated TLR7-mediated cutaneous inflammation and arthritic pathology in murine models, establishing this epigenetic modulator as a novel therapeutic target [238].

Immunometabolism has emerged as a research focus in psoriasis, with metabolic reprogramming of immune cells, offering novel therapeutic opportunities. Inhibition of Vγ4^+^γδ T cell mitochondrial translation has been demonstrated to effectively reduce erythema, desquamation, and skin thickening [239]. In psoriatic keratinocytes, protein phosphatase 6 (PP6) deficiency promotes inflammation through enhanced oxidative phosphorylation (OXPHOS), while OXPHOS inhibition significantly ameliorates disease manifestations in murine models [91]. Glutaminase 1 (GLS1)-mediated glutamine metabolism drives Th17 and γδT17 cell differentiation via the mucosa-associated lymphoid tissue lymphoma translocation protein 1 (MALT1)/c-Jun axis, and pharmacological inhibition of this pathway markedly decreases epidermal hyperplasia and inflammation in psoriatic mouse models [240]. These findings collectively suggest that targeting immunometabolic pathways represents a promising therapeutic strategy for psoriasis.

Emerging therapeutic strategies have demonstrated that engineered pyrrole-imidazole polyamides specifically bind to the binding site of c-Rel transcription factor within the IL-23p19 subunit promoter, selectively suppressing IL-23 expression and showing therapeutic efficacy in both imiquimod-induced psoriatic and experimental autoimmune uveitis murine models [241]. Cytotoxic T-lymphocyte antigen-4 (CTLA-4) serves as a negative regulator of T-cell responses through competitive binding with CD80/CD86 to inhibit CD28 co-stimulatory signaling, thereby exerting protective effects against autoimmune pathogenesis [242,243,244]. CTLA4-targeting therapeutics have advanced into clinical trials for the treatment of psoriasis and juvenile idiopathic arthritis [244,245]. The synthetic peptide dNP2-ctCTLA-4, comprising a cell-penetrating peptide (dNP2) fused to CTLA-4’s cytoplasmic signaling domain, facilitates intracellular delivery of therapeutic cargo. By expanding Tregs populations, this construct significantly ameliorates psoriasiform skin inflammation [246].

Bacterial Extracts: The novel exopolysaccharide Ebosin, isolated from *Streptomyces* sp. 139, suppresses Th17 cell differentiation while enhancing regulatory Tregs proportions, exerting anti-inflammatory effects via modulation of the miR-155- *tnfaip3*-IL-17 axis and ameliorating imiquimod induced psoriatic inflammation [247]. Similarly, leukotoxin (LtxA), a naturally occurring leukocyte-targeting bacterial protein, binds lymphocyte function-associated antigen-1 (LFA-1) to inhibit hyperproliferation of activated leukocytes in psoriasis patients, demonstrating potent therapeutic efficacy in psoriatic xenograft models [248].

### 5.4. Small-Molecule Inhibitors

Significant progress in the treatment of psoriasis using small-molecule inhibitors has been made, which provides patients with more therapeutic options. RORγt is a key regulator that promotes Th17 cell differentiation and IL-17 production. Currently, RORγt is considered a novel target for psoriasis drug development. VTP-43742 showed efficacy in alleviating psoriasis symptoms in a Phase IIa clinical trial, though potential safety concerns may exist due to insufficient participant numbers [249]. Apremilast, a phosphodiesterase-4 (PDE4) inhibitor, has been approved for treating psoriasis [250]. H4 receptor antagonists exhibited both antipruritic and anti-inflammatory effects not only in psoriasis mouse models but also for the first time in human clinical trials [251]. In K14-VEGF transgenic mice, flonoltinib maleate (FM), a dual JAK2/FLT3 inhibitor, ameliorated psoriasis-like pathology by concurrently suppressing splenic Th1/Th17 cell differentiation and limiting dendritic cell infiltration [252]. Deucravacitinib, a selective tyrosine kinase 2 inhibitor, was reported to outperform both placebo and apremilast while maintaining favorable tolerability in moderate-to-severe plaque psoriasis [253].

### 5.5. Alternative Therapy

In traditional Chinese medicine (TCM), psoriasis is termed “Bai Bi” and is treated through a holistic syndrome differentiation approach, forming a multi-target, multi-pathway integrated intervention system with the therapeutic principle of combined internal and external treatments. Clinical patterns mainly include blood-heat syndrome, blood-stasis syndrome, blood-dryness syndrome, and dampness-heat syndrome, treated following principles like “clearing heat-cooling blood (qing re liang xue)”, “activating blood-resolving stasis (huo xue hua yu)”, “nourishing blood-moistening dryness (yang xue run zao)”, and “clearing heat-resolving dampness (qing re hua shi)” [254,255]. Internal treatments primarily employ classical formulas: Rhinoceros Horn & Rehmannia Decoction (Xijiao Dihuang Decoction) or Compound Qingdai Capsules/Pills for blood-heat syndrome; Peach Kernel & Carthamus Four Substances Decoction (Taohong Siwu Decoction) or Yinxie Capsules for blood-stasis syndrome; Xiaoyin Granules or Angelica Drink (Danggui Yinzi) for blood-dryness syndrome; Gentiana Drain the Liver Decoction (Longdan Xiegan Decoction) for dampness-heat syndrome. These medications work by regulating immune-inflammatory response, improving microcirculation, and suppressing keratinocyte hyperproliferation [256,257,258,259]. TCM external therapies demonstrate unique advantages, including topical agents like Indigo Naturalis Ointment (Qingdai Ointment) and Qinbai Ointment, along with non-pharmacological therapies such as filiform needling, fire needling, cupping, and moving cupping [260,261,262]. Herbal bath therapy (e.g., Liangxue Zhiyang Formula) can reduce PASI scores and improve hemorheology [263]. Fire needling and moving cupping modulate the Th17/Treg balance and reduce STAT3 activity, thereby alleviating inflammation [264]. While herbal fumigation combined with NB-UVB significantly improves efficacy rates [265,266]. For psoriasis patients with comorbid depression, TCM emphasizes the “liver stagnation-blood heat” (gan yu xue re) pathogenesis, employing liver-soothing formulas or five-element music therapy to regulate HPA axis function and monoamine neurotransmitters, improving psychological status [267,268]. However, the current limitations of TCM research on psoriasis treatment include small number of samples in randomized clinical trials, relatively weak basic research, and the need for further optimization in topical drug formulations and mechanistic studies. For example, the oral administration of Taohong Siwu Decoction (which promotes blood circulation and removes blood stasis) can be employed to improve microcirculation, while the topical application of Qinbai Ointment provides anti-inflammatory and antipruritic effects. This approach may be combined with NB-UVB phototherapy. Furthermore, on the basis of conventional treatment, the adjunct use of Xiaoyao Powder or Five-Element Music Therapy can help regulate the HPA axis and modulate neurotransmitter activity. This approach addresses both physical and psychological dimensions, thereby holistically improving the patient’s condition.

Currently, no definitive biomarker exists for the diagnosis of psoriasis. Patients with moderate-to-severe forms of the disease may be assessed using general inflammatory markers. These include cytokines, C-reactive protein (CRP), and erythrocyte sedimentation rate (ESR) [269]. Such markers are also utilized in the evaluation of other autoimmune conditions. Additionally, the genetic marker HLA-C*06 may be considered, although it is not present in all patients. Analysis of immune cells and cytokine profiles in skin lesions can help predict treatment response to different biologics, such as TNF-α, IL-17, or IL-23 inhibitors. Genetic sequencing may also identify markers linked to drug response, supporting precision medicine. Individualized treatment should follow a structured process: shared decision-making between doctors and patients based on comprehensive assessment, followed by careful monitoring of efficacy and safety to guide timely adjustments.

Future treatment strategies for psoriasis should prioritize multidisciplinary cooperation. Comprehensive management involving enhanced patient education, lifestyle modification, and integration of artificial intelligence (AI) technologies spanning diagnostic assessment, therapeutic decision-making, treatment response prediction, risk surveillance, and patient communication/education can significantly improve clinical outcomes. This approach holds promise for advancing future psoriasis management toward timely, intelligent, and efficient paradigms. By managing psoriasis and its complications in an integrated manner, the overall health and quality of life of patients can be much improved.

## 6. Prospectives

The pathogenesis and treatment of psoriasis still have many questions to be addressed, which require a step forward to analyze the pathogenesis of psoriasis, the interaction between genetic and environmental factors and mechanisms, and the excavation and translation of novel immunotherapeutic targets. For example, the use of multi-omics technology to elucidate how susceptibility genes (e.g., *HLA-C**06:02, *il23r*) affect the disease phenotype through epigenetic regulation-mediated immune response. Neuroimmune interaction is also an important research direction. Further studies will be needed to elucidate the molecular mechanisms by which stress regulates the activation and function of immune cells via HPA axis or peripheral nervous system, thereby providing a theoretical foundation for psychological interventions or neuromodulation therapies. To elucidate the mechanism of disease recurrence, efforts will be required to investigate the long-term survival mechanism of TRM cells and develop targeted CD103-CAR-T cells or small-molecule inhibitors to remove TRM cells, so as to prolong the remission period.

Although therapeutic strategies for psoriasis have advanced significantly over the past few decades (currently available biologicals and drug targets are listed in Table 4). However, the disease remains incurable and some patients develop resistance to existing therapies [270]. Consequently, the development of more precise and individualized treatment strategies represents a critical future direction. By employing genetic testing, immunophenotyping, and other advanced methodologies to identify specific immune abnormalities and genetic profiles in individual patients, tailored therapeutic regimens can be designed to maximize efficacy. Despite the demonstrated success of existing biologicals in psoriasis treatment, their limitations, including high treatment costs, potential infection risks and long-term safety concerns, remain significant barriers to broader clinical application [250].

In the field of immunotherapy, it is imperative to overcome the limitations of existing drug targets, particularly the key molecular mediators in the pathogenesis of psoriasis, though these targets still require further in-depth investigation. Future research should be encouraged to explore novel intervention strategies targeting molecular pathways such as the RORγt/NFAT1 axis [135] to address resistance to IL-17/IL-23 inhibitors in certain patient populations. Of note, small-molecule inhibitors of RORγt (e.g., VTP-43742) [249,271] and JAK inhibitors (e.g., Tofacitinib) [272] have demonstrated promising potential in clinical trials and may emerge as viable therapeutic alternatives for psoriasis in the future.

In recent years, cell-based therapies have achieved groundbreaking advances in oncology, and their potential applications in autoimmune diseases are increasingly gaining attention. Notably, an unexpected case of psoriasis remission following CD19 CAR-T cell therapy mediated through B-cell depletion has unveiled a novel therapeutic avenue for this disease [173,219]. CAR-T cell therapy may disrupt the IL-23/Th17 axis-driven pathological paradigm and restore immune homeostasis by depleting aberrantly activated B cells or modulating T-cell subsets. Moving forward, CAR-T cell therapy and other cell-based strategies (e.g., CAR-Tregs, MSCs) demonstrate significant potential as transformative treatment modalities for psoriasis, though their long-term safety and efficacy require further validation through large-scale clinical trials. MSCs and their exosomes demonstrate therapeutic potential through immunomodulation and paracrine effects; however, challenges remain in standardized production, heterogeneity control, and long-term safety. While gene-editing technologies (e.g., CRISPR/Cas9) have shown promise in murine models, the efficiency of delivery systems and off-target risks represent major barriers to clinical translation.

Future research should focus on exploring immune tolerance induction as a novel treatment approach, particularly through antigen-specific immunotherapy targeting psoriasis-associated autoantigens such as K17 [60,171] and ADAMTSL5 [51,273]. For instance, the targeting of antigen-specific T cells or B cells by blocking antigen-specific responses or inducing tolerance to self-antigens may effectively control the inflammatory response of psoriasis without compromising systemic immunity. Additionally, bacterial extracts (e.g., Ebosin) have demonstrated anti-inflammatory effects in psoriatic animal models by modulating immune responses and inflammatory pathways [247], suggesting their potential as future therapeutic options. With rapid advancements in nanotechnology and RNA interference (RNAi), biomaterial-based delivery systems and siRNA therapeutics hold considerable promise for psoriasis treatment. For instance, HMGCS1 siRNA [215] and IL-36α siRNA [218] have exhibited significant anti-inflammatory efficacy in murine models, highlighting their potential for clinical translation.

**Table 4 ijms-26-09233-t004:** Novel and Investigational Therapies for Psoriasis.

Category	Drug Name	Target/Mechanism	Approval Status	Efficacy Profile	Adverse Effects/Potential Risks	Notes
IL-17 monoclonal antibody	Secukinumab [209]	IL-17A neutralization; blocks IL-17A signaling	Approved (US, EU, CN, etc.)	Psoriasis Area and Severity Index (PASI) 90 response: 80–90%; rapid lesion clearance	Injection-site reactions, candidiasis	Requires infection monitoring
	Vunakizumab (China-developed) [274]	Blocks IL-17A signaling	Approved (China)	Annual dosing: 14 injections; PASI 100 response: >70%	Low infection risk	Cost-effective domestic innovator
	Bimekizumab [210]	Dual IL-17A/F inhibition	Approved (EU, UK; US pending)	Superior to IL-17A monotherapy (higher PASI 100 rates)	Oral candidiasis, diarrhea	Synergistic dual-target action
IL-23 monoclonal antibody	Guselkumab [275]	IL-23p19 blockade	Approved (US, EU, CN, etc.)	Q8W dosing; PASI 90 response >80%	Mild injection-site reactions, low Tuberculosis risk	Sustained long-term remission
	Ustekinumab [206]	Dual IL-12/23p40 blockade; inhibits Th1/Th17 pathways	Approved (Global)	Long-term disease control in moderate-to-severe plaque psoriasis	Respiratory infections	First dual-target biologic for IL-12 and IL-23
TNF-α monoclonal antibody	Adalimumab [276]	TNF-α blockade	Approved (Global)	PASI 75 response: 70–80% in moderate-to-severe cases	Tuberculosis reactivation, potential malignancy risk	Preferred for psoriatic arthritis comorbidity
IL-36 monoclonal antibody	Spesolimab [212]	IL-36 blockade	Approved (US, EU, CN)	Rapid control of generalized pustular psoriasis (GPP)	Infections, infusion reactions	First-in-class IL-36 pathway inhibitor
Small-Molecule Inhibitors	Apremilast [250]	DE4 inhibitor; reduces proinflammatory cytokines	Approved in multiple countries	Suitable for mild-to-moderate cases; oral administration	Diarrhea, nausea, weight loss	Superior safety profile compared to traditional immunosuppressants
	Tofacitinib [272]	JAK1/3 inhibition; blocks JAK-STAT signaling	Approved for PsA in multiple countries	Oral administration; rapid relief of articular symptoms	Infection risk, thromboembolic events	Requires long-term safety monitoring
	VTP-43742/PF-06763809 [249,277]	RORγt inhibition; reduces IL-17 production	Clinical trials	Novel Th17 pathway suppression with promising efficacy	Good tolerability/safety pending	First-in-class RORγt inhibitors
Cell Therapies	CD19 CAR-T [173]	CD19-targeted B-cell depletion	Case reports only	Complete psoriasis remission sustained	CRS, B-cell aplasia-related infections	Unclear mechanism; target optimization needed
	CAR-Tregs [221]	Engineered Tregs for enhanced immune suppression	Preclinical studies	Effective in experimental autoimmune models (exploratory for psoriasis)	Technical complexity, graft rejection risks	Potential tolerance-restoring approach
	Umbilical/Adipose MSCs [227,228,229]	Immunomodulation (paracrine effects)	Clinical trials	Good safety profile, preliminary evidence of sustained improvement in some patients	Transient fever, infusion-related reactions	Requires stringent quality control
TCM Therapy	Compound Indigo Capsule [278]	Multi-target modulation	Approved (China)	Significant improvement in erythema and infiltration	Diarrhea, abdominal pain	Contraindicated in pregnancy
miRNA	miR-340 siRNA [214]	Downregulates IL-17A expression via RNA interference	Preclinical studies	Attenuates inflammation in murine models	Low delivery efficiency, poor stability	Requires nanocarrier optimization
Immunometabolic Modulator	GLS1 Inhibitor [240]	Glutamine metabolism blockade (Th17 differentiation)	Preclinical studies	Markedly improves imiquimod (IMQ)-induced psoriasiform dermatitis in mice	Unknown	Targets metabolic reprogramming, avoids direct immunosuppression
Epigenetic Modulator	KAT8 Inhibitor [238]	Reduces H4K16ac (suppresses CXCL2/CCL3)	Preclinical studies	Ameliorates IMQ-induced murine model symptoms	Unknown	N/A
Other Investigative Drugs	CYnLIP (Nanocarrier) [218]	Co-delivers IL-36α siRNA + erlotinib	Preclinical studies	Significantly reduces murine PASI scores	Human safety unverified	Combines gene therapy + chemical drug
	Ebosin (*Streptomyces* exopolysaccharide) [247]	Inhibits Th17 differentiation; modulates miR-155-TNFAIP3-IL-17 axis	Preclinical studies	Attenuates inflammation in murine models	Unclear toxicity profile	Natural product with multi-pathway modulation
	Antibody-Nanoparticle Conjugate [104]	Neutrophil-specific delivery of anti-inflammatory payload	Preclinical studies	Reduces systemic toxicity; site-specific action	Preclinical safety pending	Precision delivery technology prototype

Due to space limitation, not all investigational psoriasis therapies are shown.

Immunometabolic studies have revealed the pivotal role of metabolic reprogramming (e.g., OXPHOS, glutaminolysis) in psoriasis pathogenesis. Targeting these pathways (e.g., via GLS1 inhibitors) may represent a promising therapeutic strategy. Epigenetic regulation (e.g., KAT8 inhibition) offers precise intervention targets by modulating inflammatory gene expression, though its tissue specificity and safety profile require further investigation.

TCM demonstrates unique advantages in psoriasis treatment through multi-target regulation (e.g., IL-17 inhibition, Th17/Treg balance modulation), particularly for mild-to-moderate cases or in synergistic combination with Western medicine. However, the lack of high-quality evidence-based medical data and standardized protocols remains a bottleneck for its global adoption. Future development requires integration with modern technologies to advance precision medicine in integrative TCM-Western approaches.

In summary, the therapeutic landscape for psoriasis holds immense promise, with future research continuing to explore precise, safe, and effective treatment strategies. Interdisciplinary collaboration and technological innovation will facilitate the translation of the research results from bench to clinical applications, ultimately advancing psoriasis therapy toward individualized, precision-based and integrated approaches. While significant challenges remain, the continuous scientific and technological advancements will ultimately render the cure for psoriasis an achievable reality.

## Figures and Tables

**Figure 1 ijms-26-09233-f001:**
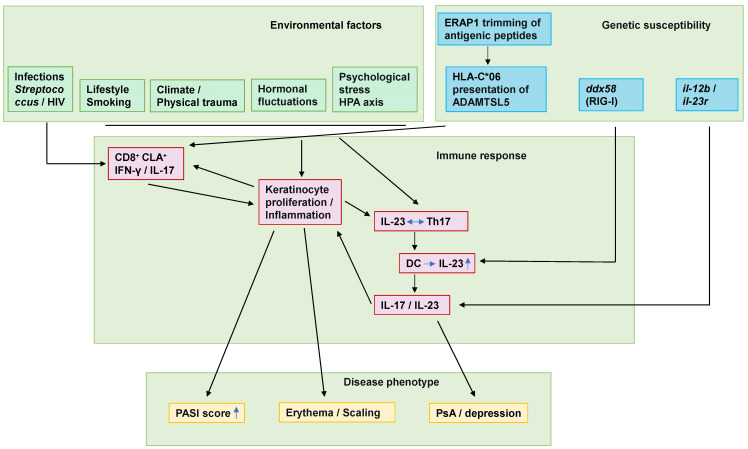
Schematic diagram of genetic and environmental factors driving immune activation in psoriasis. Environmental triggers (including infections, lifestyle factors, climate, hormonal fluctuations, and psychological stress) activate the immune system. Key genetic factors involve antigen presentation pathways (e.g., ERAP1 trimming antigenic peptides for presentation by HLA-C*06), innate immune sensors (e.g., RIG-I), and cytokine signaling (e.g., IL-12p40/IL-23R). This leads to immune activation characterized by CD8^+^ CLA^+^ T cell responses, IFN-γ and IL-17 production, and IL-23/Th17 axis amplification. Subsequent keratinocyte proliferation and inflammation manifest clinically as increased disease severity, elevated Psoriasis Area and Severity Index (PASI) score, erythema, scaling, and the development of comorbidities such as psoriatic arthritis (PsA) or depression.

**Figure 2 ijms-26-09233-f002:**
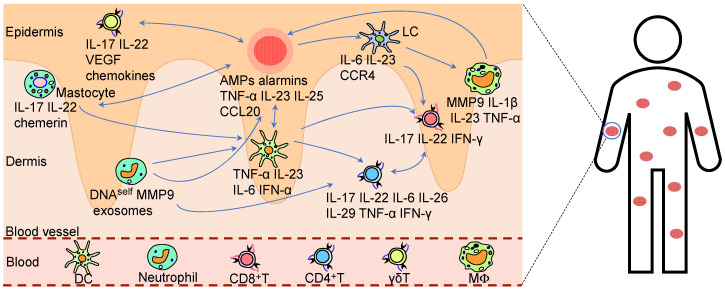
Schematic diagram of dysregulated immune responses in psoriasis. In genetically predisposed individuals, skin injury or infectious triggers induce neutrophil infiltration, releasing neutrophil extracellular traps (NETs) that activate the TLRs-IFN-α pathway in dendritic cells (DCs), while exosomes and matrix metalloproteinase-9 (MMP9) further enhance the secretion of cytokines by keratinocytes (KCs). KCs release antimicrobial peptides (AMPs), cytokines, and alarmins, recruiting other immune cells to the inflammatory site. Additionally, DCs and Langerhans cells (LCs) drive T-cell differentiation via IL-23. CD4^+^ Th17 cells, CD8^+^ Tc17 cells, and γδ T cells collectively produce cytokines such as IL-17, further stimulating KCs proliferation and proinflammatory cytokine release, establishing a self-amplifying immune feedback loop. Moreover, tumor necrosis factor (TNF)-α and IL-1β secreted by macrophages and KCs synergistically amplify inflammation. These dysregulated immune responses lead to profound immune cell infiltration, excessive production of cytokines and chemokines, and KCs pathological hyperproliferation, resulting in characteristic psoriatic lesions featuring hyperkeratosis and parakeratosis.

**Table 2 ijms-26-09233-t002:** Differential Diagnosis of Psoriasis.

Differential Diagnosis Item	Distinctive Features	Commonly Affected Areas
Atopic dermatitis	Lesions may have ill-defined borders, but lichenified plaques with relatively clear borders may develop due to scratching	Flexural surfaces of the body
Seborrheic dermatitis	Dandruff presents as greasy, yellow scales or crusts; the degree of underlying erythema varies and is often more pronounced on the face	Sebaceous gland-rich regions
Tinea capitis	Often accompanied by hair loss and broken hairs; inflammatory reactions (e.g., pustules, abscesses) may be present; fungal microscopy is positive	Scalp, hairline
Secondary syphilis	Diverse rash morphology (can mimic various skin diseases); copper-red papules or macules; positive syphilis serological tests	Trunk, limbs
Lichen planus	Violaceous, polygonal, flat-topped papules with white reticulated streaks (Wickham’s striae) on the surface	Wrists, forearms, ankles, oral mucosa, genitalia
Chronic eczema	Lesions have ill-defined borders; often presents with lichenification (skin thickening and accentuated skin markings) and hyperpigmentation; intense pruritus	Variable sites
Rheumatoid arthritis	Rheumatoid factor (RF) is often positive; typically involves small joints (wrists, metacarpophalangeal, proximal interphalangeal joints) symmetrically; prolonged morning stiffness; absence of psoriatic skin lesions and nail changes	Small joints
Candidiasis/Fungal infection	Well-demarcated red patches, potentially with satellite lesions; scales are non-silvery; positive fungal microscopy or culture	Skin folds
Pityriasis rosea	Herald patch: a larger initial lesion appears 1–2 weeks before the generalized eruption; lesions align with skin cleavage lines (“Christmas-tree” distribution); collarette scaling.	Trunk, proximal limbs
Cutaneous T-cell lymphoma	Early stages may present with solitary or multiple erythematous to dusky red patches/plaques, with thin, non-silvery scales; may progress to plaques and nodules; diagnosis is confirmed by pathological biopsy.	Can occur anywhere on the body

## Data Availability

All data in this paper were acquired from reference literatures (https://pubmed.ncbi.nlm.nih.gov/ and https://www.cnki.net/). No new data were created in this paper.

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
