# Peer review of "Advances in Psoriasis Research: Decoding Immune Circuits and Developing Novel Therapies"

_ijms, 2025, doi:10.3390/ijms26189233_

Round 1
Reviewer 1 Report
Comments and Suggestions for Authors
Comment 1: The abstract provides a solid overview but remains too broad and lacks clear scope, failing to indicate whether the review emphasises immunology, therapeutics, or precision medicine. It omits discussion of key therapeutic classes such as biologics and JAK inhibitors, which limits its relevance to current clinical practice. The novelty of the review is also unclear, as it does not specify what unique perspective or contribution it offers compared with existing literature. While systemic comorbidities are briefly mentioned, their importance is underdeveloped, and the mention of microbiome modulation feels superficial without linking it to psoriasis pathogenesis. Finally, the reference to individualised treatment is vague, with no concrete examples of biomarkers or strategies, reducing the impact of the precision medicine angle.
Comment 2: The introduction provides a comprehensive overview of psoriasis, including its clinical manifestations, systemic comorbidities, pathogenic mechanisms, genetic predispositions, and environmental triggers. While this breadth demonstrates a strong command of the literature, it is overly detailed for an introduction and risks overshadowing the main focus of the review. Key aspects such as the IL-23/IL-17 axis, microbial involvement, and lifestyle factors are well described but could be presented more concisely. Importantly, the introduction lacks a clear statement of novelty and does not sufficiently highlight the unique perspective or contribution of this review. To strengthen the manuscript, the section would benefit from trimming treatment details, refining the scope to align more closely with the stated objectives, and explicitly clarifying what new insights the review seeks to provide.
Comment 3: Although section 2 on epidemiology and disease burden covers relevant points but can be improved. First, the prevalence data are inconsistent and lack contextualisation, the wide range (0.09%–11.4%) is presented without discussion of differences in study design, population demographics, or diagnostic criteria, which risks misinterpretation, and the claim that rising prevalence is linked to antimicrobial misuse and environmental pollution is speculative and not sufficiently substantiated with mechanistic or longitudinal evidence. The UK study cited on pollution exposure is interesting but feels isolated, without reference to comparable findings globally, limiting its generalisability. The link to DALYs is mentioned but not integrated into a coherent discussion of socioeconomic burden. The psychosocial impact is important, yet the section could benefit from a deeper exploration of how these mental health outcomes interact with disease severity and treatment access.
Commnet 4: Section 3 is informative and comprehensive, demonstrating depth of clinical knowledge, but could benefit from structured synthesis. The detailed descriptions of each psoriasis subtype are valuable, but the information would be clearer if presented in a summary table comparing subtypes, prevalence, key clinical features, and severity. While associations such as nail involvement with psoriatic arthritis are noted, other clinically relevant correlations across subtypes are underdeveloped and could be highlighted. The brief discussion of histopathology and imaging could similarly benefit from a concise table summarising pathogenomic versus supportive features, and the differential diagnosis section would be more accessible if organised into a table outlining mimickers, distinguishing features, and affected regions.
Comment 5: Section 4 sub-sections, particularly innate and adaptive immunity, could benefit from summary tables or schematics illustrating key cell types, cytokines, and interactions, making the complex information more accessible. While the discussion of genetic and environmental triggers is comprehensive, the interplay between these factors and immune activation could be clarified with integrative diagrams or flowcharts.
Comment 6: Section 5 is a thorough and up-to-date review of psoriasis management, well-cited and scientifically rigorous. The manuscript could benefit from better integration of traditional Chinese medicine with modern therapeutic strategies to highlight potential complementary effects. Additionally, the level of mechanistic detail is inconsistent across sections; conventional systemic therapies are described more superficially compared to advanced therapies such as CAR-T cells or siRNA-based approaches. Finally, minor typographical and formatting issues, including inconsistent hyphenation, spacing, and sentence breaks, should be addressed to improve readability and polish.
Comments on the Quality of English LanguageThe manuscript is generally understandable and uses appropriate scientific terminology; however, the text would benefit from careful language refinement. Several sentences are overly long and complex, occasionally affecting clarity. Minor grammatical errors, inconsistent tenses, and punctuation issues are present, and some concepts are repeated unnecessarily. Streamlining repetitive statements, simplifying complex sentences, and ensuring consistent terminology would enhance readability and overall presentation.
Author Response
Point-by-point replies to the comments and suggestions from the reviewers
Comment 1: The abstract provides a solid overview but remains too broad and lacks clear scope, failing to indicate whether the review emphasises immunology, therapeutics, or precision medicine. It omits discussion of key therapeutic classes such as biologics and JAK inhibitors, which limits its relevance to current clinical practice. The novelty of the review is also unclear, as it does not specify what unique perspective or contribution it offers compared with existing literature. While systemic comorbidities are briefly mentioned, their importance is underdeveloped, and the mention of microbiome modulation feels superficial without linking it to psoriasis pathogenesis. Finally, the reference to individualised treatment is vague, with no concrete examples of biomarkers or strategies, reducing the impact of the precision medicine angle.
Response: Thank you very much for the comment and critics. We revised the abstract as required. We realized that systemic comorbidities associated with psoriasis significantly reduce patients' quality of life, posing multiple threats to their living standards, socioeconomic status, and overall safety. (Please see lines 15-17 and 79-82 in the revised manuscript). We have emphasized the focus of this review on the abstract and expanded the discussion on key therapeutic categories such as biologics and JAK inhibitors. (Please see lines 23-30, 19-22 in the revised manuscript). As psoriasis currently lacks disease-specific biomarkers, from moderate to severe cases, indicators such as cytokines, C-reactive protein, and erythrocyte sedimentation rate are usually measured. These indexes are also commonly assessed in other inflammatory or autoimmune diseases. The genetic marker HLA-C*06, while somewhat specific, is not present in all patients. We have added the information regarding psoriasis biomarkers and personalized treatment to the manuscript. (Please see lines 758-768 in the revised manuscript). In the Pathogenesis section, we delineated the role of microbial factors (Pathogen associated molecular patterns, PAMPs) in the pathogenesis of psoriasis. Given the need for conciseness in the abstract, this aspect was only be briefly mentioned therein. (Please see lines 236-240 in the revised manuscript).
Comment 2: The introduction provides a comprehensive overview of psoriasis, including its clinical manifestations, systemic comorbidities, pathogenic mechanisms, genetic predispositions, and environmental triggers. While this breadth demonstrates a strong command of the literature, it is overly detailed for an introduction and risks overshadowing the main focus of the review. Key aspects such as the IL-23/IL-17 axis, microbial involvement, and lifestyle factors are well described but could be presented more concisely. Importantly, the introduction lacks a clear statement of novelty and does not sufficiently highlight the unique perspective or contribution of this review. To strengthen the manuscript, the section would benefit from trimming treatment details, refining the scope to align more closely with the stated objectives, and explicitly clarifying what new insights the review seeks to provide.
Response: Thank you very much for the comments and valuable suggestion. In the manuscript, we reviewed the novel treatments using miRNAs, CAR-T related therapy, gene therapy, exosomes etc., and extended it to alternative therapy. However, we found it difficult to trim the treatment section because we tried to cover as many valuable literatures as possible to let the readers learn the current research advances (please see lines 536-776). We have streamlined certain details in the introduction and clarified the insights provided by this review. (Please see lines 61-63 in the revised manuscript).
Comment 3: Although section 2 on epidemiology and disease burden covers relevant points but can be improved. First, the prevalence data are inconsistent and lack contextualisation, the wide range (0.09%–11.4%) is presented without discussion of differences in study design, population demographics, or diagnostic criteria, which risks misinterpretation, and the claim that rising prevalence is linked to antimicrobial misuse and environmental pollution is speculative and not sufficiently substantiated with mechanistic or longitudinal evidence. The UK study cited on pollution exposure is interesting but feels isolated, without reference to comparable findings globally, limiting its generalisability. The link to DALYs is mentioned but not integrated into a coherent discussion of socioeconomic burden. The psychosocial impact is important, yet the section could benefit from a deeper exploration of how these mental health outcomes interact with disease severity and treatment access.
Response: Thank you very much for the valuable suggestion. We have added the prevalence data with background analyses covering diagnostic variations and geographical distributions. (Please see lines 65-69 in the revised manuscript). The proposed links between antibiotic misuse, environmental pollution, and increased incidence of psoriasis remain speculative. There is still a relative scarcity of high-quality epidemiological studies directly establishing antibiotic overuse as a causative factor for rising psoriasis rates. Nevertheless, this has become a growing area of research interest, although the underlying biological mechanisms are not yet fully understood. Current basic research suggests that antibiotic abuse may disrupt gut microbiota homeostasis and alter the immunomodulatory functions of microbial metabolites, thereby potentially exacerbating the development of psoriasis in genetically susceptible individuals. In addition, several studies have reported that environmental pollutants may trigger or worsen the disease. Relevant references supporting these associations have been added and appropriately cited in the manuscript. (Please see lines 69-75 in the revised manuscript). Due to the absence of comprehensive global statistics on Disability-Adjusted Life Years (DALYs) related to psoriasis, with existing studies being restricted to specific populations in certain countries, all mentions of DALYs have been removed from the manuscript. Emphasis has been laid on the particular importance of psychological counseling and therapy for patients with psoriasis comorbid with depression. (Please see lines 85-87 in the revised manuscript).
Commnet 4: Section 3 is informative and comprehensive, demonstrating depth of clinical knowledge, but could benefit from structured synthesis. The detailed descriptions of each psoriasis subtype are valuable, but the information would be clearer if presented in a summary table comparing subtypes, prevalence, key clinical features, and severity. While associations such as nail involvement with psoriatic arthritis are noted, other clinically relevant correlations across subtypes are underdeveloped and could be highlighted. The brief discussion of histopathology and imaging could similarly benefit from a concise table summarising pathogenomic versus supportive features, and the differential diagnosis section would be more accessible if organised into a table outlining mimickers, distinguishing features, and affected regions.
Response: Thank you very much for the constructive comments. We have incorporated tables detailing psoriasis subtypes and differential diagnoses in the manuscript as suggested. (Please see table1 and table2 in the revised manuscript). We have emphasized the clinical correlations among the various subtypes. (Please see lines 102-103, 113-114, 117-119, 121-124, 128-131, 133-135 in the revised manuscript). Considering that tables take much more space than the text, we decided not to use tables in the histopathology and imaging sections.
Comment 5: Section 4 sub-sections, particularly innate and adaptive immunity, could benefit from summary tables or schematics illustrating key cell types, cytokines, and interactions, making the complex information more accessible. While the discussion of genetic and environmental triggers is comprehensive, the interplay between these factors and immune activation could be clarified with integrative diagrams or flowcharts.
Response: Thank you very much for the valuable suggestion. The interaction between genetic and environmental triggers and immune activation has been illustrated schematically. (Please see figure1 in the revised manuscript). We have summarized the key cell types and cytokine interactions in a schematic diagram. (Please see figure2 in the revised manuscript).
Comment 6: Section 5 is a thorough and up-to-date review of psoriasis management, well-cited and scientifically rigorous. The manuscript could benefit from better integration of traditional Chinese medicine with modern therapeutic strategies to highlight potential complementary effects. Additionally, the level of mechanistic detail is inconsistent across sections; conventional systemic therapies are described more superficially compared to advanced therapies such as CAR-T cells or siRNA-based approaches. Finally, minor typographical and formatting issues, including inconsistent hyphenation, spacing, and sentence breaks, should be addressed to improve readability and polish.
Response: Thank you very much for the comment. We have incorporated strategies that integrate traditional Chinese medicine with modern therapeutic approaches. (Please see lines 749-757 in the revised manuscript). We have included the mechanisms of action of conventional systemic therapies and provided a more in-depth description of these treatments. (Please see lines 501-516, 514-517, 528-531 in the revised manuscript). We have addressed issues related to layout and formatting, including inconsistencies in hyphen usage, spacing, and sentence breaks. (Please see lines 46-47, 94-97, 99-100, 105-106, 153, 161-164, 165-167, 221-224, 276, 279-281, 287-290, 331-333, 396, 457, 569-571, 625-628, 740-742, 791-794 in the revised manuscript).
Comment 7: The manuscript is generally understandable and uses appropriate scientific terminology; however, the text would benefit from careful language refinement. Several sentences are overly long and complex, occasionally affecting clarity. Minor grammatical errors, inconsistent tenses, and punctuation issues are present, and some concepts are repeated unnecessarily. Streamlining repetitive statements, simplifying complex sentences, and ensuring consistent terminology would enhance readability and overall presentation.
Response: Thank you very much for the comment. We have asked a native English-speaker and professional editor to check for language issues such grammatical errors, miswording, and long sentences, removing redundant concepts such as the Koebner phenomenon, simplified complex sentences, corrected inconsistent tenses and punctuation, and ensured terminological consistency throughout the manuscript. (Please see lines 46-47, 94-97, 99-100, 105-106, 153, 161-164, 165-167, 221-224, 276, 279-281, 287-290, 331-333, 396, 457, 569-571, 625-628, 740-742, 791-794 in the revised manuscript).
Reviewer 2 Report
Comments and Suggestions for Authors
Thank you for the opportunity to review this thoughtful and ambitious narrative on the pathobiology and therapeutics of psoriasis. The manuscript is clear, well-organized, and successfully connects fundamental immunology, particularly the IL-23/IL-17 axis, to both established and emerging treatments. I especially appreciated the comprehensive scope across modalities, including conventional systemics, phototherapy, biologics, small molecules, and cellular or gene-adjacent strategies. The effort to frame investigational areas (exosomes, CAR-T/Tregs) for a clinical audience is commendable. Additionally, the cautious approach taken in discussing traditional medicine regarding evidence quality is beneficial for readers outside of that literature.
To strengthen accuracy and enhance reader usability, I offer the following suggestions:
1. Update Regulatory Status: Please refresh the regulatory status throughout the therapy tables and relevant text. In Table 2, the row for bimekizumab currently states “Approved (EU, UK; US pending).” Given the pace of global decisions, I recommend updating the approvals using a compact regional style: USA, EU/UK, Canada, Japan, China, with years included where space allows.
2. Specify Efficacy Data: Where efficacy is summarized, replacing broad ranges with trial- and timepoint-specific numbers will prevent inadvertent overstatement. For example, the secukinumab entry lists “PASI 90 response: 80–90%,” but it would be clearer to present a specific figure with a timepoint (week 16 or 48) and, a brief footnote. This will allow readers to quickly compare outcomes across classes and agents. The literature supporting precise, timepoint-specific PASI 75/90/100 outcomes is available.
3. Clarify Small-Molecule Safety: In the small-molecule section, the RORγt row currently states “Good tolerability/safety pending,” which feels vague given observations from early programs. I suggest clarifying the specific safety signals seen in early clinical development so that readers understand why enthusiasm is tempered.
4. Refine Epidemiology Paragraph: The epidemiology paragraph would benefit from two small refinements for clarity. The sentence reporting a worldwide prevalence “ranging from 11.4% to 0.09%” reads abruptly and in reverse order; presenting the low-to-high direction and pairing it with a simple global anchor will better orient readers.
5. Adjust Wording for Precision: Two brief wording adjustments can prevent misinterpretation. In the introduction, “long-term use may be associated with increased risks of drug resistance and infections” would be more precise if stated as secondary loss of response (immunogenicity or mechanistic escape) and infection risk. This will align with clinical experience and avoids implying antimicrobial-style resistance.
6. Expectations for Cellular Therapy Data: The manuscript already conveys evidence levels for emerging modalities, which I applaud. In the cellular therapy section, consider adding a leading sentence noting that human data are limited to case reports or very small cohorts, with known risks (infection risk from B-cell aplasia, cytokine release syndromes) and unresolved long-term outcomes.
Thank you again for your manuscript and great work. I applaud the authors for their efforts and look forward to seeing the final revisions.
Author Response

(The authors gave the same response as above.)

Reviewer 3 Report
Comments and Suggestions for Authors
This review article provides a comprehensive and timely overview of psoriasis, summarizing epidemiology, clinical presentation, pathophysiology, and current as well as emerging therapies. The manuscript is ambitious in scope and well-structured, covering recent developments in immune mechanisms, biologics, small-molecule inhibitors, cellular therapies, and alternative approaches. The breadth of information is valuable, particularly for readers seeking an up-to-date synthesis of both classical and cutting-edge aspects of psoriasis research.
The main strength of the paper is its extensive coverage of immunological pathways, highlighting the central role of the IL-23/IL-17 axis while also addressing contributions from keratinocytes, dendritic cells, neutrophils, γδ T cells, and B cells. The section on novel therapies, including CAR-T cells, Treg-based strategies, and exosome therapy, is particularly relevant as it provides perspectives that go beyond standard biologics.
At the same time, the manuscript would benefit from some refinements. The text is very detailed but occasionally overwhelming, making it challenging for readers to follow the main messages. Condensing certain sections, avoiding redundancy, and improving transitions would enhance readability. The therapeutic section is particularly long, and a more synthetic presentation (e.g., summary tables highlighting mechanisms, efficacy, and safety) would help readers grasp the key points. In addition, while the manuscript reviews many novel approaches, the critical appraisal is limited. A more balanced discussion of the limitations, risks, and current stage of development of these therapies would improve the paper’s scientific value. Finally, although the English is generally understandable, the manuscript would benefit from professional language editing to correct minor grammatical errors and improve fluency.
Author Response

(The authors gave the same response as above.)

Round 2
Reviewer 1 Report
Comments and Suggestions for Authors
The revision have addressed major deficiencies in the submitted draft.
Reviewer 2 Report
Comments and Suggestions for Authors
The revised manuscript is substantially improved. The authors have clearly and thoughtfully addressed the prior comments, resulting in a clearer, more focused, and well-organized review. I appreciate the significant effort invested in these revisions. I have no further substantive comments and recommend acceptance as is.